# INVARIANCE VS. ROBUSTNESS OF NEURAL NETWORKS

## ABSTRACT

Neural networks achieve state of the art accuracy on many standard datasets used in image classification. One would like to design neural network models whose performance is not affected by natural (or non-malicious) geometric transformations as well as adversarial (or malicious) perturbations. Previous work has studied generalization to natural geometric transformations (e.g., rotations) as *invariance*, and generalization to adversarial perturbations as *robustness*. In this paper, we examine the interplay between *invariance* and *robustness*. We empirically study the following two cases: (a) change in adversarial robustness as we improve only the invariance using equivariant models and training augmentation, (b) change in invariance as we improve only the adversarial robustness using adversarial training. We observe that the rotation invariance of equivariant models (StdCNNs and GCNNs) improves by training augmentation with progressively larger rotations but while doing so, their adversarial robustness drops progressively and this drop is very significant on MNIST. As a plausible explanation for this phenomenon we observe that the average perturbation distance of the test points to the decision boundary decreases as the model learns larger and larger rotations. We take adversarially trained LeNet and ResNet models which have good $\ell_\infty$ adversarial robustness on MNIST and CIFAR-10, respectively, and observe that adversarially training them with progressively larger norms results in a progressive drop in their rotation invariance profiles. On these adversarially trained models we observe that the change in the rate of invariance is marginal for small rotations. As a plausible explanation for this phenomenon, we show empirically that the principal components of adversarial perturbations and the principal components of perturbations given by small rotations are nearly orthogonal.

## 1 INTRODUCTION

Neural networks achieve state of the art accuracy on several standard datasets used in image classification. However, their performance in the wild depends on how well they can handle natural or non-adversarial transformations of input seen in real-world data as well as known deliberate, adversarial attacks created to fool the model.

Natural or non-adversarial transformations seen in real-world images include translations, rotations, and scaling. Convolutional Neural Networks (CNNs) are translation-invariant or shift-invariant by design. Invariance to other symmetries, and especially rotations, have received much attention recently, e.g., Harmonic Networks (H-Nets) by Worrall et al. (2016), cyclic slicing and pooling by Dieleman et al. (2016), Transformation-Invariant Pooling (TI-Pooling) by Laptev et al. (2016), Group-equivariant Convolutional Neural Networks (GCNNs) by Cohen & Welling (2016), Steerable CNNs by Cohen & Welling (2017), Deep Rotation Equivariant Networks (DREN) by Li et al. (2017), Rotation Equivariant Vector Field Networks (RotEqNet) by Marcos et al. (2017), Polar Transformer Networks (PTN) by Esteves et al. (2018). For a given symmetry group $G$, a $G$-equivariant network learns a representation or feature map at every intermediate layer such that any transformation $g \in G$ applied to an input corresponds to an equivalent transformation of its representations. Any model can improve its invariance to a given group of symmetries through sufficient training augmentation. Equivariant models use efficient weight sharing and require smaller sample complexity to achieve better invariance. Equivariant models such as CNNs and GCNNs too generalize well to progressively larger random rotations, but only when their training data is augmented similarly.

Adversarial attacks on neural network models are certain, deliberate changes to inputs that fool a highly accurate model but are unlikely to fool humans. Given any neural network model, Szegedy et al. (2013) show how to change the pixel values of images only slightly so that the change is almost imperceptible to human eye but makes highly accurate models misclassify. Szegedy et al. (2013) find these adversarial pixel-wise perturbations of small magnitude by maximizing the prediction error of a given model using box-constrained L-BFGS. Goodfellow et al. (2015) propose Fast Gradient Sign Method (FGSM) that applies the adversarial perturbation as $x' = x + \epsilon \, \text{sign} \left( \nabla_x J(\theta, x, y) \right)$, where $x$ is the input, $y$ represents the target, $\theta$ represents the model parameters, and $J(\theta, x, y)$ is the loss function used to train the network. Moreover, Goodfellow et al. (2015) propose adversarial training, or training augmented with points $(x', y)$, as a way to improve adversarial robustness of a model.

Subsequent work has introduced multi-step variants of FGSM, notably, an iterative method by Kurakin et al. (2017) and Projected Gradient Descent (PGD) attack by Madry et al. (2018). Given any model, these attacks produce adversarial perturbation for every test image $x$ from a small $\ell_\infty$-ball around it, namely, each pixel value $x_i$ is perturbed within $[x_i - \epsilon, x_i + \epsilon]$. PGD attack does so by solving an inner optimization by projected gradient descent over $\ell_\infty$-ball of radius $\epsilon$ around $x$ to approximate the optimal perturbation. Adversarial training with PGD perturbations improves the adversarial robustness of models, and it is one of the best known defenses to make models robust to perturbations of bounded $\ell_\infty$ norm on MNIST and CIFAR-10 datasets, as shown in Madry et al. (2018) and Athalye et al. (2018).

Recent work has looked at simultaneous robustness to multiple adversarial attacks. Engstrom et al. (2017) show that adversarial training with PGD makes CNNs robust against perturbations of bounded $\ell_\infty$ norm but an adversarially chosen combination of a small rotation and a translation can still fool these models nevertheless. Recent work of Schott et al. (2019) shows that PGD adversarial training is a good defense against perturbations of bounded $\ell_\infty$ norm but can be broken with adversarial perturbations of small $\ell_0$ or $\ell_2$ norm that are also imperceptible to humans or have little semantic meaning for humans. Schott et al. (2019) show how to build models for MNIST dataset that are simultaneously robust to perturbations of small $\ell_0$, $\ell_2$ and $\ell_\infty$ norms.

## 1.1 Problem formulation and our results

Let $f$ be a neural network classifier trained on a training set of labeled images. The accuracy of $f$ is the fraction of test inputs $x$ for which the predicted label $f(x)$ matches the true label $y$. For a given transformation $T$ of the image space, $f$ is said to be $T$-invariant if the predicted label remains unchanged after the transformation $T$ for all inputs, i.e., $f(Tx) = f(x)$, for all inputs $x$. If $f$ is $T$-invariant and $f(x + a) \neq f(x)$, for some input $x$ and its adversarial perturbation $a$ with $\|a\|_p \leq \epsilon$, then by invariance $f(Tx + Ta) = f(x + a) \neq f(x) = f(Tx)$. Let $T$ be a translation, rotation, or more generally abstracted as a permutation of coordinates in Tramèr & Boneh (2019), then $\|Ta\|_p = \|a\|_p \leq \epsilon$. Hence, $Ta$ is an adversarial perturbation for input $Tx$ of small $\ell_p$ norm. When a change of variables maps $x$ to $Tx$ by a permutation of coordinates, then the gradient and $T$ operators can be swapped. In other words, $\text{grad}(f)$ at $Tx$ is the same as $T$ applied to $\text{grad}(f)$ at $x$. This gives a 1-1 correspondence between FGSM (and PGD) perturbations of $x$ and $Tx$, respectively, with $\ell_p$ norm at most $\epsilon$. As a corollary, the $\ell_p$ fooling rate for any $T$-invariant classifier $f$ on the transformed data $\{Tx \; : \; x \in \mathcal{X}\}$ must be equal to its $\ell_p$ fooling rate on the original data $\mathcal{X}$.

A subtlety kicks in when $f$ is not truly invariant, that is, $f(Tx) = f(x)$, for most inputs $x$ but not all $x$. Define the *rate of invariance* of a classifier $f$ to a transformation $T$ as the fraction of test images whose predicted labels after transformation $T$ remains unchanged, i.e., $f(Tx) = f(x)$. For a class of transformations, e.g., rotations upto degree $[-\theta^\circ, +\theta^\circ]$, we define the *rate of invariance* as the average rate of invariance over transformations $T$ in this class. The rate of invariance is 100% iff the model $f$ is truly invariant. When $f$ is not truly invariant, the interplay between the invariance under transformations and robustness under adversarial perturbations of small $\ell_p$-norm is subtle. This interplay is exactly what we investigate.

In this paper, we study neural network models and the simultaneous interplay between their rate of invariance for random rotations between $[-\theta^\circ, +\theta^\circ]$, and their adversarial robustness to pixel-wise perturbations of $\ell_\infty$ norm at most $\epsilon$.

Measuring the robustness of a model to adversarial perturbations of $\ell_p$ norm at most $\epsilon$ is NP-hard as shown in Katz et al. (2017) and Sinha et al. (2018). Athalye et al. (2018) compare most of the

known adversarial attacks to argue that PGD is among the strongest. Therefore, we use accuracy on adversarially perturbed test data using PGD attack as a proxy for adversarial robustness.

Unlike previous studies by Engstrom et al. (2017) and Tramèr & Boneh (2019), we do not fix the magnitude of pixel-wise adversarial perturbations (e.g., say $\epsilon = 0.3$) nor limit ourselves to small rotations up to $\pm30°$. Another important difference is we consider random rotations instead of adversarial rotations. We compute the rate of invariance of a given model on inputs rotated by a random angle between $[-\theta°, +\theta°]$, for $\theta$ varying in the range $[0, 180]$. Similarly, we normalize the underlying dataset, and compute the accuracy of a given model on adversarially perturbed test inputs by PGD and FGSM adversarial attacks of $\ell_\infty$ norm at most $\epsilon$, for $\epsilon$ varying in the range $[0, 1]$.

We empirically study the following: (a) change in $\ell_\infty$ adversarial robustness as we improve only the rate of rotation invariance using training augmentation with progressively larger rotations, (b) change in invariance as we improve only adversarial robustness using PGD adversarial training with progressively larger $\ell_\infty$-norm of pixel-wise perturbations.

We study equivariant models, StdCNNs and GCNNs, as well as LeNet and ResNet model from Madry et al. (2018). Equivariant models, especially GCNNs, when trained with random rotation augmentations come very close to being truly rotation invariant; see Cohen & Welling (2016). Adversarially trained LeNet and ResNet models with PGD are among the best known $\ell_\infty$ adversarially robust models on MNIST and CIFAR-10, respectively, as shown in Madry et al. (2018), and reconfirmed by Athalye et al. (2018). In other words, these models essentially represent the two separate solutions known currently for achieving invariance and adversarial robustness, respectively.

Our two main observations are as follows.

(i) Equivariant models (StdCNNs and GCNNs) progressively improve their rate of rotation invariance when their training is augmented with progressively larger random rotations but while doing so, their $\ell_\infty$ adversarial robustness drops progressively. This drop or trade-off is very significant on MNIST.

(ii) Adversarially trained LeNet and ResNet models from Madry et al. (2018) trained using progressively larger $\ell_\infty$-norm attacks improve their adversarial robustness but while doing so, their rate of invariance to random rotations upto $\pm\theta°$ drops progressively. On these adversarially trained models we observe that the change in the rate of invariance is marginal for small rotations.

We empirically observe that the average distance of the test points to the decision boundary reduces when models are trained with larger rotations. This could explain observation (i). We empirically observe that adversarial perturbations are nearly orthogonal to perturbations caused by small rotations of test points. This could explain the marginal change in the rate of invariance for small rotations, mentioned in (ii).

**Related Work** Schott et al. (2019) study simultaneous robustness to adversarial perturbations of small $\ell_0$, $\ell_2$, and $\ell_\infty$-norm. Tramèr & Boneh (2019) show an impossibility result by exhibiting data distribution where no model can have substantially better-than-random accuracy for binary classification simultaneously against both $\ell_\infty$ and $\ell_1$ perturbations, and also against both $\ell_\infty$ perturbations and spatial perturbations, given by an adversarial permutation of coordinates that models an adversarially chosen combination of a small rotation and a small translation. They show an empirical validation of this claim on MNIST and CIFAR-10 datasets for simultaneous robustness against $\ell_\infty$ adversarial perturbation and an adversarially chosen combination of translation upto $\pm3$ pixels and rotation upto $\pm30°$. Intuitively and theoretically, it has been argued in Engstrom et al. (2019) and Tramèr & Boneh (2019) that *small, adversarial pixel-wise perturbations* and *small, adversarial geometric transformations* are essentially dissimilar attacks focusing on different features, due to which a simultaneous solution to both may be inherently difficult. They do not postulate any gradual trade-off between invariance and robustness. They do not consider group-equivariant models such as GCNNs, a natural choice for invariance to geometric transformations.

## 2 ROTATION INVARIANCE VS. $\ell_\infty$ ADVERSARIAL ROBUSTNESS

In this section, we present our main result about the interplay between rotation invariance and $\ell_\infty$ adversarial robustness of models on MNIST and CIFAR-10 data. In Subsection 2.1, we take GCNN

models from Cohen & Welling (2016) and study their rotation invariance and $\ell_\infty$ adversarial robustness, as we train them with random rotations of progressively larger degree. In Subsection 2.2, we take LeNet and ResNet models from Madry et al. (2018) and study their rotation invariance and $\ell_\infty$ adversarial robustness, as we do PGD adversarial training with progressively larger $\ell_\infty$ norms.

We present our experimental results as rotation invariance profiles and adversarial robustness profiles, as explained below.

Rotation invariance means that the predicted labels of an image and any of its rotations should be the same. Since most datasets are centered, we restrict our attention to rotations about the center of each image. We quantify rotation invariance by measuring the rate of invariance or the fraction of test images whose predicted label remains the same after rotation by a random angle between $[-\theta^\circ, \theta^\circ]$. As $\theta$ varies from 0 to 180, we plot this as the *rotation invariance profile* of a given model.

Adversarial robustness means that the predicted labels of an image and its adversarial perturbation should be the same. The $\ell_\infty$ adversarial robustness of a given model to a fixed adversarial attack (e.g., FGSM, PGD) and a fixed $\ell_\infty$ norm (say, $0 \le \epsilon \le 1$) is quantified by the accuracy of the given model on perturbed test data with an adversarial perturbation of $\ell_\infty$ norm at most $\epsilon$ generated by the attack. The resulting plot of the accuracy of the model as $\epsilon$ varies from 0 to 1 is the *robustness profile* of the given model.

**Convention used in the legends of our figures:** We use the following convention in the legends of some plots. A coloured line labeled $A/B$ indicates that the training data is augmented with random rotations from $[-A^\circ, A^\circ]$ and the test data is augmented with random rotations from $[-B^\circ, B^\circ]$. If $A$ (resp. $B$) is zero it means the training data (resp. test data) is unrotated. If the model is trained with random rotations from $[-A^\circ, A^\circ]$ and the test data is randomly rotated with varying $B$ to draw the plot, we only mention $A$ and not $B$, which is self-explanatory.

Equivariant models (e.g., GCNNs, H-Nets, PTNs, RotEqNets) are designed to be robust to rotations through clever weight sharing. However, models such as GCNNs and H-Nets are not robust to rotations if their training data is not sufficiently augmented. This appears to be folklore so we do not elaborate on this in the main paper. See Appendix E.

## 2.1 Effect of rotation invariance on $l_\infty$ adversarial robustness

For any fixed $\theta \in [0, 180]$, we take an equivariant model, namely, StdCNN or GCNN, and augment its training data by random rotations from $[-\theta^\circ, +\theta^\circ]$. Figure 1(left), Figure 2(left) shows how the robustness profile of StdCNN change on MNIST and CIFAR-10 respectively, as we increase the degree $\theta$ used in training augmentation of the model. We use PGD adversarial attack for MNIST and CIFAR-10. Figure 1(right) and Figure 2(right) show the rotation invariance profile of the same models on MNIST, CIFAR-10 respectively.

The black line in Figure 1 (left), shows that the adversarial robustness of a StdCNN which is trained to handle rotations up to $\pm 180$ degrees on MNIST, drops by more than $50\%$, even when the $\epsilon$ budget for PGD attack on unrotated MNIST is only $0.1$. The black line in Figure 1 (right), shows this models rotation invariance profile - this model is invariant to larger rotations in the test data. This can be contrasted with the model depicted by the red line - this StdCNN is trained to handle rotations up to 60 degrees. The rotation invariance profile of this model is below that of the model depicted by the black line and so it is not invariant to large rotations. However this model can handle adversarial $\ell_\infty$-perturbations up to $0.3$ on unrotated data, with an accuracy more than $80\%$ - this can be seen from the red line in Figure 1 (left).

From these plots it is clear that *the rotation invariance of these models improves by training augmentation but at the cost of their adversarial robustness, indicating a trade-off between invariance to rotations and adversarial robustness.* The above observations, of there being a trade-off between handling larger rotations with training augmentation and handling larger adversarial perturbations is seen in GCNNs also. This can be seen from Figures 28 and Figure 29. The plots are very similar to what we observe with StdCNNs.

We have additional experiments in Appendix B, where we look at the adversarial accuracy of the above models when the test data is also augmented with random rotations. The conclusions are similar. Unlike the invariance profiles, our adversarial robustness profiles do not start at $100\%$

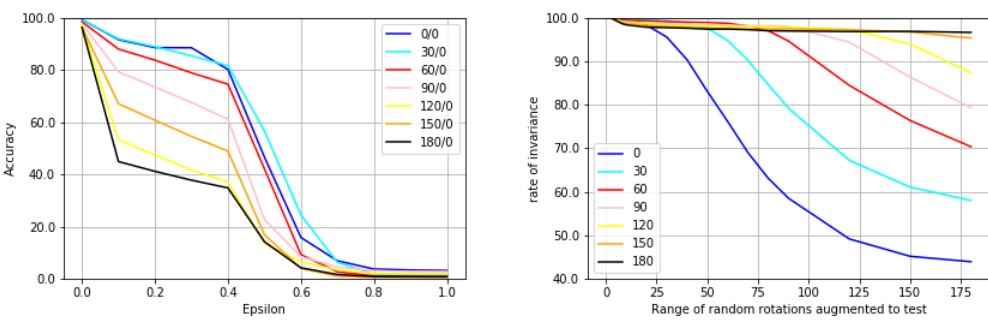

Figure 1: On MNIST, StdCNN trained with varying random rotations in $[-\theta°, \theta°]$ range. (left) Robustness profile, (right) Rotation invariance profile.

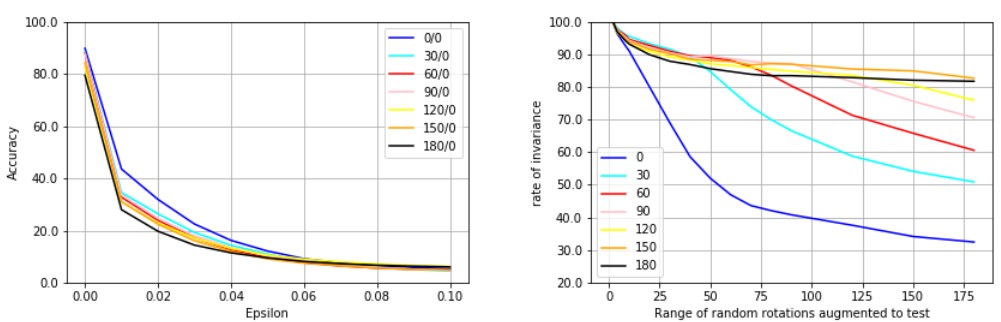

Figure 2: On CIFAR-10, StdCNN/VGG16 trained with varying random rotations in $[-\theta°, \theta°]$ range. (left) Robustness profile, (right) Rotation invariance profile.

because their starting point is the natural accuracy on unperturbed test data. The natural accuracy on unperturbed test data is known to drop after adversarial training, as noted by Tsipras et al. (2019). In Appendix F, we plot (1 - fooling rate) instead of accuracy, so that the adversarial robustness profiles also start with 100%. The fooling rate is defined as the fraction of test images whose predicted label changes after adversarial perturbation.

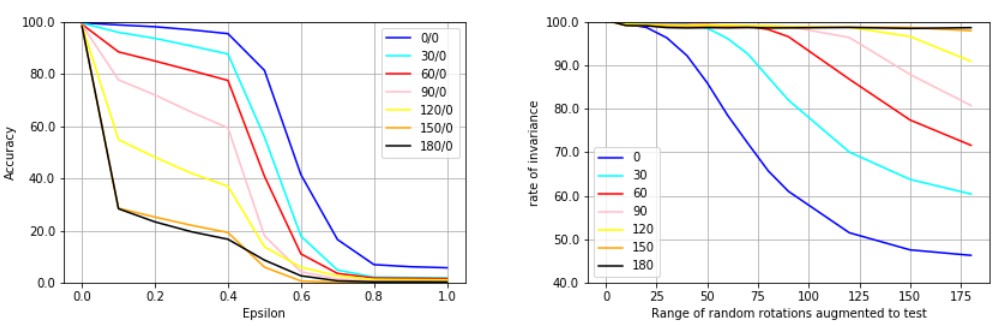

Figure 3: On MNIST, GCNN trained with varying random rotations in $[-\theta°, \theta°]$ range. (left) Robustness profile, (right) Rotation invariance profile.

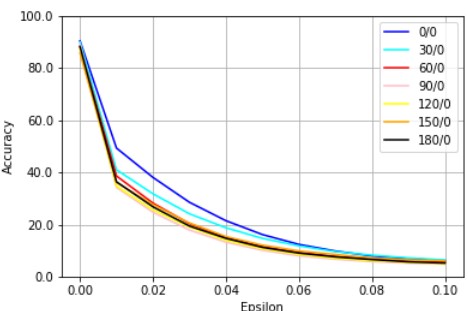 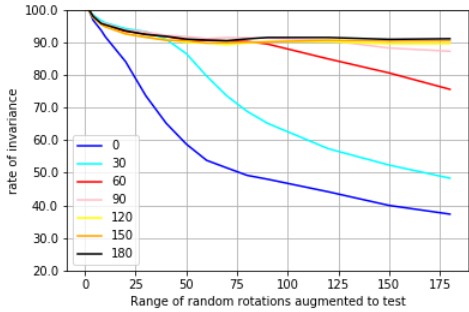

Figure 4: On CIFAR-10, GCNN/VGG16 trained with varying random rotations in $[-\theta°, \theta°]$ range. (left) Robustness profile, (right) Rotation invariance profile.

## 2.2 EFFECT OF $\ell_\infty$ ADVERSARIAL TRAINING ON ROTATION INVARIANCE

The most common approach to improve adversarial robustness is adversarial training, i.e., training the model on adversarially perturbed training data. Adversarial training with PGD attack is one of the strongest known defenses on MNIST and CIFAR-10 datasets, see Athalye et al. (2018).

For any fixed $\epsilon \in [0, 1]$ we adversarially train our models, LeNet and ResNet, as given in Madry et al. (2018) with PGD adversarial perturbations with $\ell_\infty$ budget $\epsilon$. Similar to Madry et al. (2018) we use the LeNet model for MNIST and the ResNet model for CIFAR-10. We then plot their rotation invariance profiles and their robustness profiles. Each colored line in Figure 5 and Figure 6 corresponds to a model adversarially trained with a different value of $\epsilon$.

On MNIST, adversarial training with PGD with larger $\epsilon$ results in a drop in the invariance profile of LeNet based model - in Figure 5 (left), the yellow line (PGD with $\epsilon = 0.4$) is below the light blue line (PGD with $\epsilon = 0.1$). This is true for the ResNet based model too on CIFAR-10, as can be seen from Figure 6 (left). In other words, *adversarial training with progressively larger $\epsilon$ leads to the drop in the rate of invariance on the test data.*

To complete this picture we plot the robustness profile curves of the LeNet and ResNet based model for MNIST and CIFAR-10, respectively. It is known that as these models are trained with PGD using larger $\epsilon$ budget their adversarial robustness increases. The robustness profile curves of the LeNet model trained with larger PGD budget dominates the robustness profile curve of the same model trained with a smaller PGD budget - the red line in Figure 5 (right), dominates the light blue line. This is true of the ResNet based model too, as can be seen from Figure 6 (right).

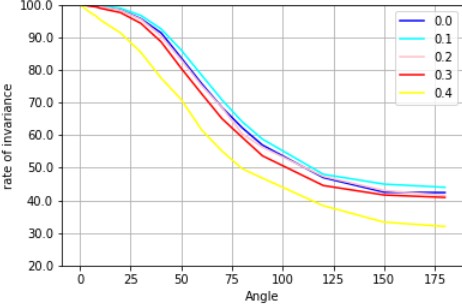 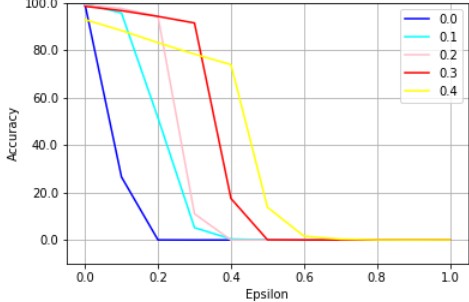

Figure 5: For PGD adversarially trained LeNet based model from Madry et al. (2018) on MNIST (left) Rotation invariance profile, (right) Robustness profile. Different colored lines represent models adversarially trained with different $\ell_\infty$ budgets $\epsilon \in [0, 1]$.

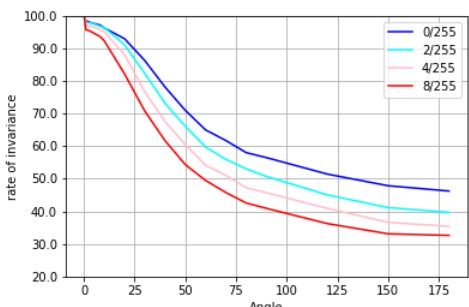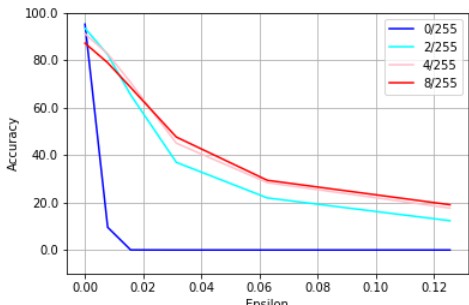

Figure 6: For PGD adversarially trained ResNet based model from Madry et al. (2018) on CIFAR-10 (left) Rotation invariance profile, (right) Robustness profile. Different colored lines represent models adversarially trained with different $\ell_\infty$ budgets $\epsilon \in [0, 1]$.

## 3 DECISION BOUNDARY DISTANCE AND ORTHOGONALITY

In order to understand what could be the possible reasons behind our empirical observations in Section 2.1 and Section 2.2 we perform more experiments.

### 3.1 AVERAGE PERTURBATION DISTANCE TO THE BOUNDARY.

For each test image, adversarial attacks find perturbations of the test point with small $\ell_\infty$ norm that would change the prediction of the given model. Most adversarial attacks do so by finding the directions in which the loss function of the model changes the most. In order to explain why these networks become vulnerable to pixel-wise attacks as they learn more rotations, we see how the distance of the test points to the decision boundary changes as the networks learn larger rotations. This is abstractly depicted in Figure 7 where we show the distance of a test point $x_0$ to the boundary $D_0$ (resp. $D_{180}$) when the model is trained with zero (resp. $\pm 180°$) rotations.

We use the $L_2$ attack vectors obtained by DeepFool (Moosavi-Dezfooli et al., 2016) for the datapoints under attack. We take the norm of this attack vector as an approximation to the shortest distance of the test point to the decision boundary. For each of the test points we collect the perturbation vectors given by DeepFool attack and report the average perturbation distance. We plot this average distance as the datasets are augmented with larger rotations.

Our experiments show that as the networks learn larger rotations with augmentation, the average perturbation distance falls. So as (symmetric) networks become invariant to rotations, they are more vulnerable to pixel-wise attacks.

The plots in Figures 8, 9, 10, 11 show this for StdCNNs and GCNNs on MNIST and CIFAR-10. To make our point we plot the accuracy of these networks and also the average perturbation distance of the test points alongside in one figure. The blue line in Figure 8(left) shows the accuracy of a StdCNN on MNIST when both the training data and test data are augmented with $\theta$, as $\theta$ ranges from 0 to 180. The green line in Figure 8(left) shows the accuracy of the StdCNN model when the train is augmented with random rotations upto $\theta$, and the test is augmented with rotations upto $\theta$ and is also perturbed with PGD of $\ell_\infty$ norm 0.3. The red line shows the accuracy when the test is not augmented with rotations but is PGD perturbed with $\ell_\infty$ norm 0.3.

The red line Figure 8(right) shows the average perturbation distance of the unrotated test points when the network is trained with rotations upto $\theta$. The green line shows the average perturbation distance of test points which are augmented with rotations upto $\theta$ - this is about 5 when $\theta$ is $0°$ (the point on the $y$-axis where the curves begin). As the network is trained with random rotations up to $180°$ the average perturbation distance of the augmented test drops below 3.5. Figure 8(left) shows that that the PGD accuracy has dropped from around 85% for the network at $0°$ to 30% at $180°$ (the corresponding green line on the left). The fact that the red line is above the green line is also reflected in Figure 14 in Appendix C. When the test is perturbed by PGD, the accuracy of the

StdCNN with training data augmented with rotations is better when the test is not augmented with rotations than if the test were also augmented with rotations.

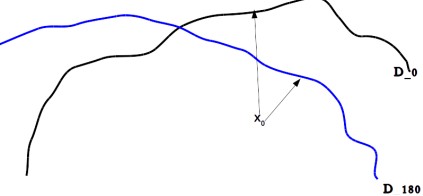

Figure 7: Distance of point $x_0$ to decision boundary $D_{180}$ obtained by augmenting training set with random rotations in range $[-180°, 180°]$ is different compared to the decision boundary $D_0$ obtained with no training augmentation.

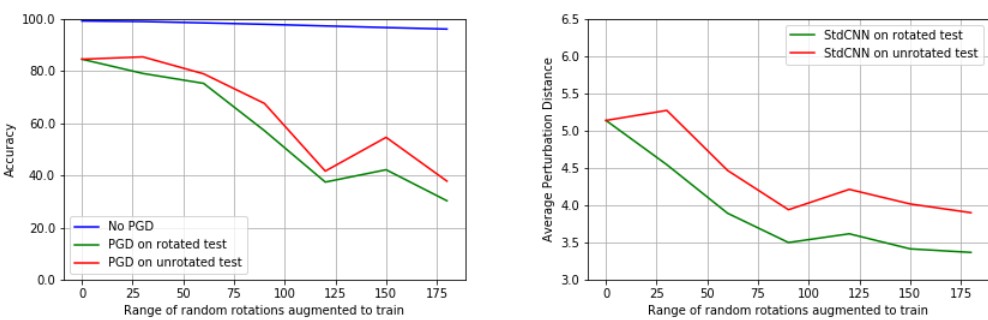

Figure 8: Accuracy of StdCNN on MNIST with/without PGD ($\epsilon = 0.3$), on rotated and unrotated test. Train/test if augmented are with random rotations in $[-\theta°, \theta°]$. (left) Accuracy, (right) Avg. Perturbation Distance.

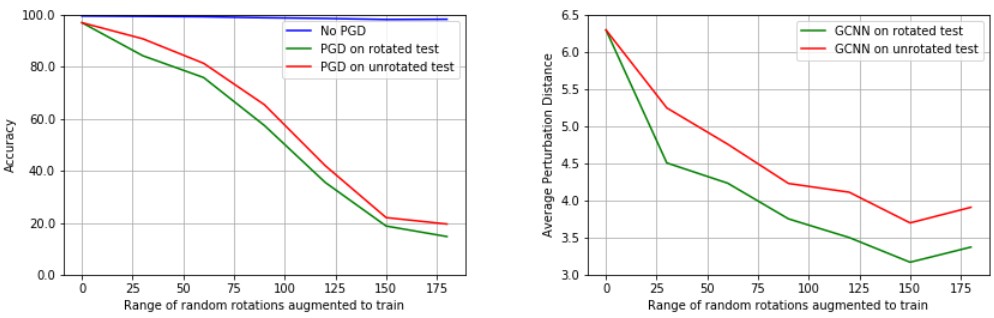

Figure 9: Accuracy of GCNNs on MNIST with/without PGD ($\epsilon = 0.3$) on rotated and unrotated test. Train/test if augmented are with random rotations in $[-\theta°, \theta°]$. (left) Accuracy, (right) Avg. Perturbation Distance.

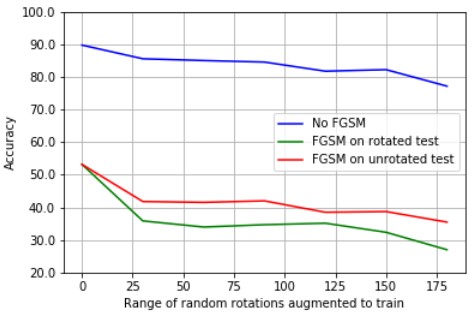 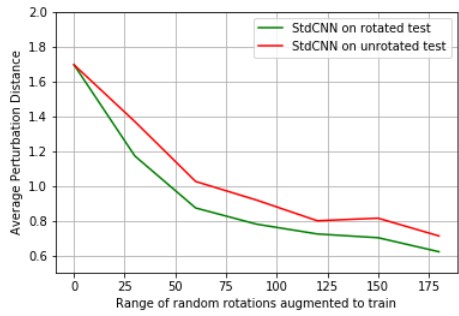

Figure 10: Accuracy of StdCNNs/VGG16 on CIFAR-10, with/without FGSM ($\epsilon = 0.01$) on rotated and unrotated test with $\epsilon = 0.01$. Train/test if augmented are with random rotations in $[-\theta°, \theta°]$. (left) Accuracy, (right) Avg. Perturbation Distance.

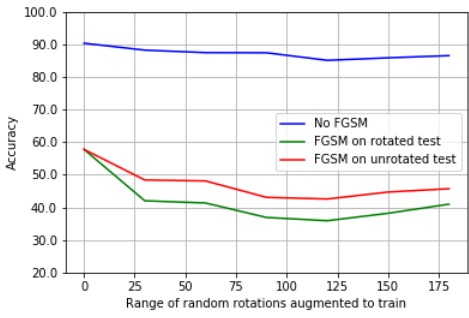 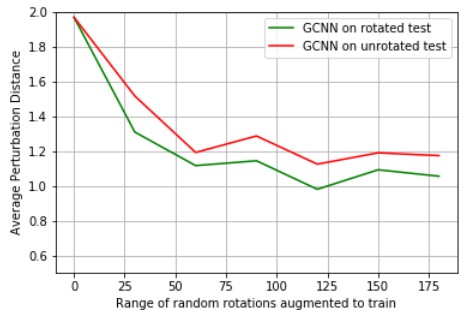

Figure 11: Accuracy of GCNNs/VGG16 on CIFAR-10, with/without FGSM ($\epsilon = 0.01$) on rotated and unrotated test. Train/test if augmented are with random rotations in $[-\theta°, \theta°]$. (left) Accuracy, (right) Avg. Perturbation Distance.

## 3.2 ORTHOGONALITY OF ADVERSARIAL PERTURBATIONS & INVARIANT ROTATIONAL PERTURBATIONS

In general we expect neural networks to classify $x$ and a small rotation $x'$ of $x$, identically. This would be particularly so if the dataset has some inherent rotations and the labels remain the same under such rotations. Writing $x' = x + \Delta(x)$, we say $\Delta(x)$ is an invariant rotational perturbation for $x$. For a network trained to be equivariant to rotations, this rotational perturbation of $x$ could well be orthogonal to an adversarial perturbation of $x$, since an adversarial perturbation changes the label whereas an invariant rotational perturbation does not change the label. We empirically show that this is the case for MNIST and CIFAR-10. For each test input we find its PGD perturbation and stack these adversarial perturbations as rows of an adversarial matrix. We find the top 5 right singular vectors of this adversarial matrix. We rotate each input by $4°/2°$ for MNIST/CIFAR-10 and subtract the original image from it. We stack such invariant rotational perturbations as rows of an invariance matrix. We find the top 5 right singular vectors of this invariance matrix. We empirically observe that the principal angles between these two 5-dimensional subspaces are close to $90°$. For completeness we recall the definition of principal angles in Appendix D.

Table 1 gives the principal angles of MNIST trained on LeNet and Table 2 gives the principal angles for CIFAR-10 trained on ResNet as given by Madry et al. (2018).

Table 1: MNIST: Principal angles between Top-5 SVD-subspace of PGD attack directions of test points with $\ell_\infty$ norm $\epsilon$ and Top-5 SVD-subspace of the difference directions between a test point and its $4°$ rotation.

| $\epsilon$ | 1 | 2 | 3 | 4 | 5 |
|---|---|---|---|---|---|
| 0.0 | 89.48 | 89.03 | 86.91 | 84.79 | 80.97 |
| 0.1 | 89.93 | 87.97 | 86.15 | 81.62 | 76.99 |
| 0.2 | 89.56 | 88.00 | 85.83 | 82.29 | 80.73 |
| 0.3 | 89.75 | 88.49 | 87.75 | 84.96 | 80.45 |
| 0.4 | 89.33 | 88.55 | 86.86 | 85.80 | 82.13 |

Table 2: CIFAR-10: Principal angles between Top-5 SVD-subspace of PGD attack directions of test points with $\ell_\infty$ norm $\epsilon$ and Top-5 SVD-subspace of the difference directions between a test point and its $2°$ rotation.

| $\epsilon$ | 1 | 2 | 3 | 4 | 5 |
|---|---|---|---|---|---|
| 0/255 | 89.99 | 89.40 | 88.27 | 86.98 | 85.99 |
| 2/255 | 89.76 | 88.76 | 88.31 | 87.15 | 86.33 |
| 4/255 | 89.78 | 89.02 | 88.30 | 86.78 | 86.41 |
| 8/255 | 89.69 | 89.13 | 88.11 | 87.25 | 86.42 |

## 4 CONCLUSION

We observe that as equivariant models (StdCNNs and GCNNs) are trained with progressively larger rotations their rotation invariance improves but at the cost of their adversarial robustness. A plausible explanation of the first observation is that the average perturbation distance of the test points to the boundary decreases as StdCNNs and GCNNs learn larger rotations. Adversarial training with perturbations of progressively increasing norms improves the robustness of LeNet and ResNet models, but with a resulting drop in their rate of invariance. On these adversarially trained models we observe that the change in the rate of invariance is marginal for small rotations. As a plausible explanation for this phenomenon, we show empirically that the principal components of adversarial perturbations and the principal components of perturbations given by small rotations are nearly orthogonal.

**Acknowledgement**  We thank an anonymous reviewer for suggestions that improved the presentation of our invariance profile plots and the invariance-robustness trade-off significantly.

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

Table 3: Architectures used for the MNIST and Fashion MNIST experiments

| Standard CNN | GCNN |
|---|---|
| Conv(10,3,3) + Relu | P4ConvZ2(10,3,3) + Relu |
| Conv(10,3,3) + Relu | P4ConvP4(10,3,3) + Relu |
| Max Pooling(2,2) | Group Spatial Max Pooling(2,2) |
| Conv(20,3,3) + Relu | P4ConvP4(20,3,3) + Relu |
| Conv(20,3,3) + Relu | P4ConvP4(20,3,3) + Relu |
| Max Pooling(2,2) | Group Spatial Max Pooling(2,2) |
| FC(50) + Relu | FC(50) + Relu |
| Dropout(0.5) | Dropout(0.5) |
| FC(10) + Softmax | FC(10) + Softmax |

## APPENDIX A   DETAILS OF EXPERIMENTS

All experiments performed on neural network-based models were done using MNIST, Fashion MNIST and CIFAR-10 datasets with appropriate augmentations applied to the train/validation/test set.

**Data sets** MNIST dataset consists of $70,000$ images of $28 \times 28$ size, divided into 10 classes. $55,000$ used for training, $5,000$ for validation and $10,000$ for testing. Fashion MNIST dataset consists of $70,000$ images of $28 \times 28$ size, divided into 10 classes. $55,000$ used for training, $5,000$ for validation and $10,000$ for testing. CIFAR-10 dataset consists of $60,000$ images of $32 \times 32$ size, divided into 10 classes. $40,000$ used for training, $10,000$ for validation and $10,000$ for testing.

**Model Architectures** For the MNIST and Fashion MNIST based experiments we use the network architecture of GCNN as given in Cohen & Welling (2016). The StdCNN architecture is similar to the GCNN except that the operations are as per CNNs. Refer to Table 3 for details. H-Nets architecture is as given in Worrall et al. (2016). RotEqNet architecture is as given in Marcos et al. (2017). PTN architecture is as given in Esteves et al. (2018).

For the CIFAR-10 based experiments we use the VGG16 architecture as given in Simonyan & Zisserman (2014) and its GCNN equivalent is obtained replacing the various layer operations with equivalent GCNN operations as given in Cohen & Welling (2016). This is similar to how we obtained a GCNN architecture from StdCNN for the MNIST based experiments. Input training data was augmented with random cropping and random horizontal flips.

For the adversarial training experiments we used the LeNet based architecture for MNIST and the ResNet architecture for CIFAR-10. Both these models are exactly as given in Madry et al. (2018).

## APPENDIX B    EFFECT OF INVARIANCE ON ADVERSARIAL ROBUSTNESS AFTER TEST AUGMENTATION

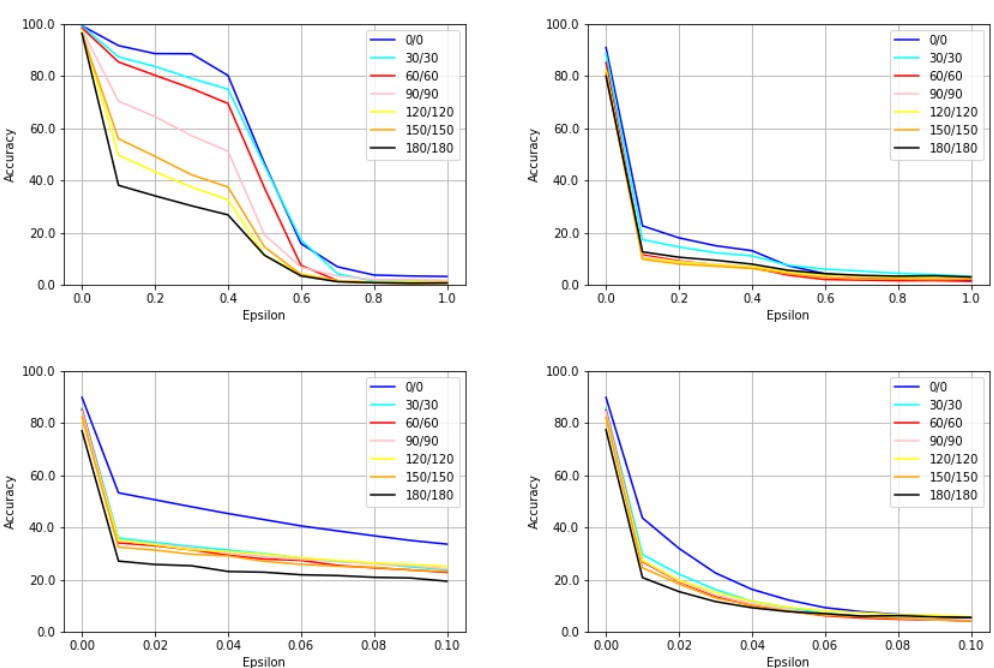

Figure 12: Robustness profile of StdCNN models. (top left) PGD attack on MNIST, (top right) PGD attack on Fashion MNIST, (bottom left) FGSM attack on CIFAR-10, (bottom right) PGD attack on CIFAR-10.

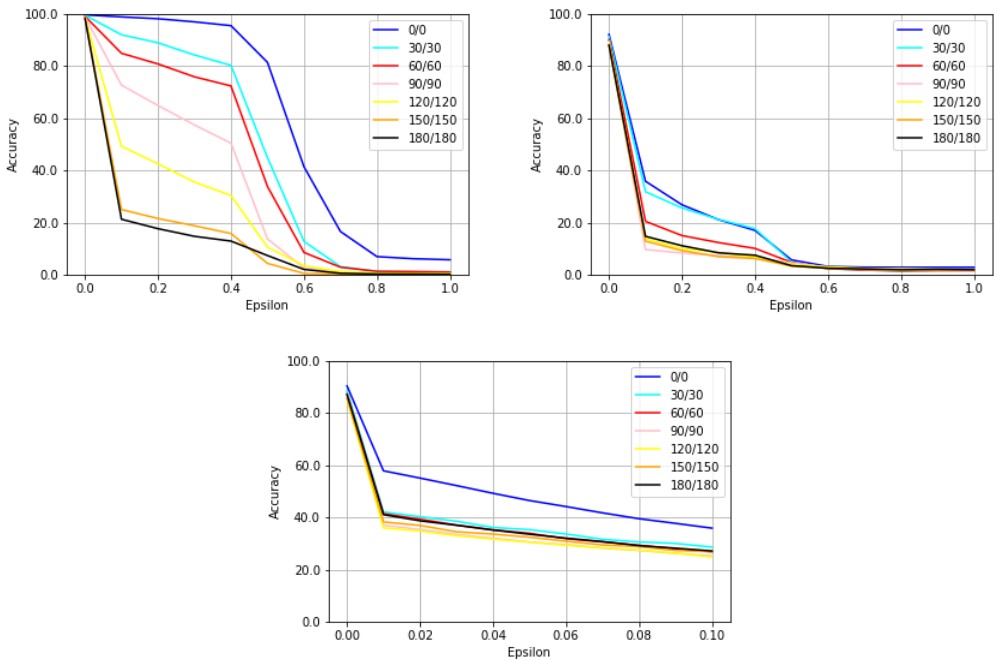

Figure 13: Robustness profile of GCNN models. (top left) PGD attack on MNIST, (top right) PGD attack on Fashion MNIST, (bottom) FGSM attack on CIFAR-10,

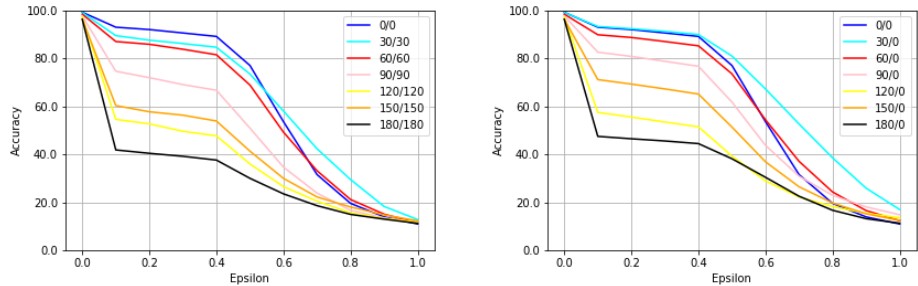

Figure 14: Robustness profile of StdCNN models on MNIST attacked with FGSM. (left) train and test augmented with $[-\theta°, \theta°]$ range (right) Only train augmented with $[-\theta°, \theta°]$ range and no test augmentation.

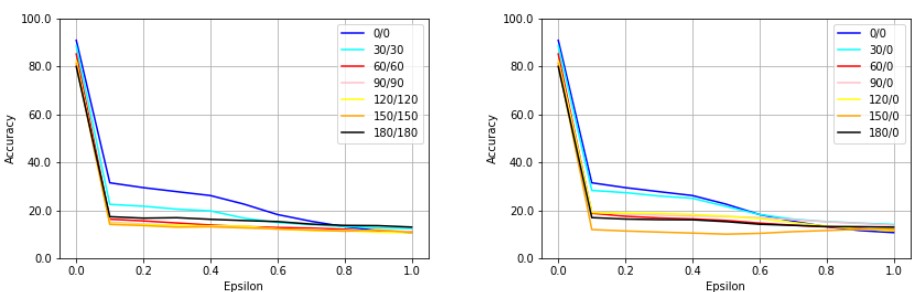

Figure 15: Robustness profile of StdCNN models on Fashion-MNIST attacked with FGSM. (left) train and test augmented with $[-\theta°, \theta°]$ range (right) Only train augmented with $[-\theta°, \theta°]$ range and no test augmentation.

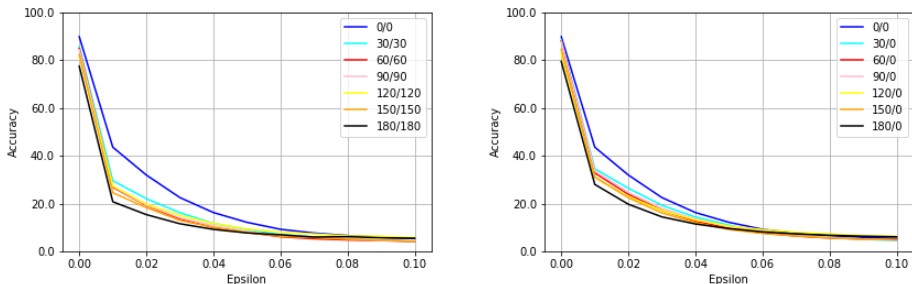

Figure 16: Robustness profile of StdCNN/VGG16 models on CIFAR-10 attacked with PGD. (left) train and test augmented with $[-\theta^\circ, \theta^\circ]$ range (right) Only train augmented with $[-\theta^\circ, \theta^\circ]$ range and no test augmentation.

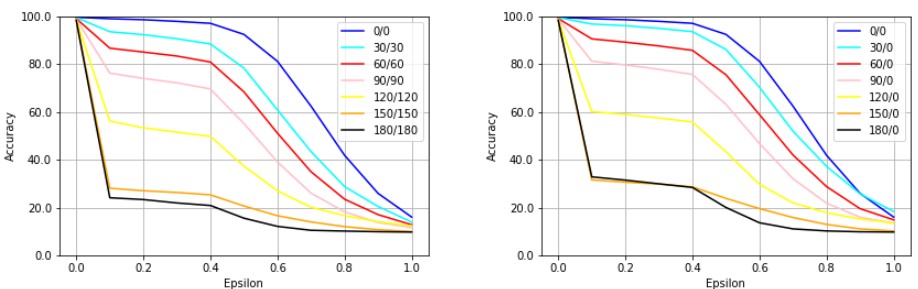

Figure 17: Robustness profile of GCNN models on MNIST attacked with FGSM. (left) train and test augmented with $[-\theta^\circ, \theta^\circ]$ range (right) Only train augmented with $[-\theta^\circ, \theta^\circ]$ range and no test augmentation.

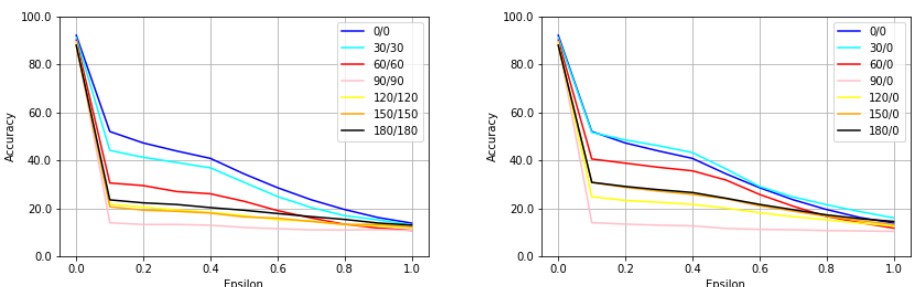

Figure 18: Robustness profile of GCNN models on Fashion-MNIST attacked with FGSM. (left) train and test augmented with $[-\theta^\circ, \theta^\circ]$ range (right) Only train augmented with $[-\theta^\circ, \theta^\circ]$ range and no test augmentation.

## APPENDIX C DECISION BOUNDARY DISTANCE, ORTHOGONALITY

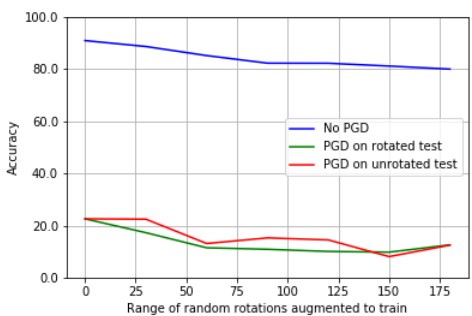 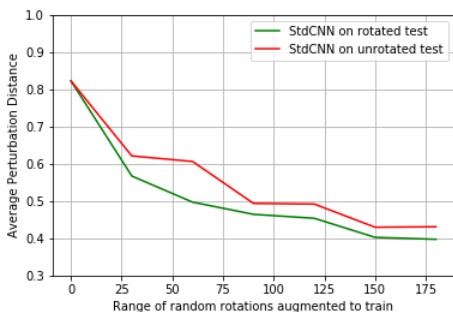

Figure 19: On Fashion MNIST, StdCNNs, Comparison of network with/without PGD on rotated and unrotated test with $\epsilon = 0.1$. Train/test if augmented are with random rotations in $[-\theta°, \theta°]$. (left) Accuracy, (right) Avg. Perturbation Distance.

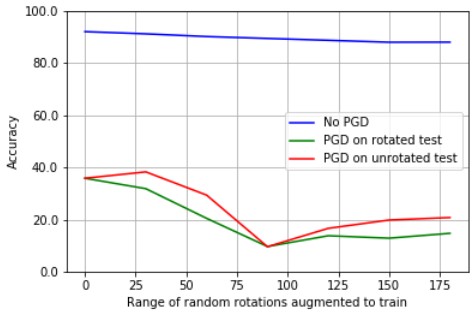 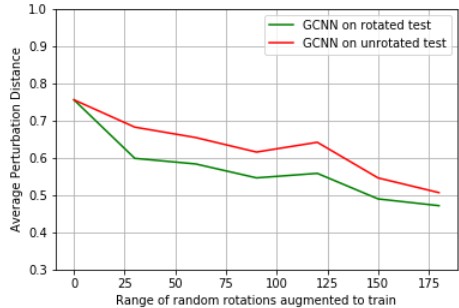

Figure 20: On Fashion MNIST, GCNNs, Comparison of network with/without PGD on rotated and unrotated test with $\epsilon = 0.1$. Train/test if augmented are with random rotations in $[-\theta°, \theta°]$. (left) Accuracy, (right) Avg. Perturbation Distance.

## APPENDIX D    PRINCIPAL ANGLES

Recall that for two subspaces $V,W$, the first principal angle is defined as the minimum angle between two unit vectors $v_1 \in V$, $w_1 \in W$. The second principal angle is the minimum angle between unit vectors $v_2 \in V, w_2 \in W$, with $v_2 \perp v_1$ and $w_2 \perp w_1$. The other principal angles are defined similarly.

## APPENDIX E    ROBUSTNESS OF ROTATION EQUIVARIANT NETWORKS TO ROTATIONS

In this appendix we confirm that many popular rotation equivariant models are not robust to rotations if their training data is not sufficiently augmented with rotations. We do this by plotting their accuracy when the test data is augmented with rotations but the training data is not augmented with rotations of the same range. We also show empirically that their accuracy improves by training with a small sample of data augmented with random rotations. We observe that (a) Rotation equivariant networks are robust to only small degrees of rotations away from the ones present in the training data - this was studied only for CNNs in Weiler et al. (2018), (b) applying data augmentation during training increases their accuracy even when the test data is augmented with rotations. (c) Rotation equivariant networks achieve state of the art results with smaller sample size for training.

### E.1    ROBUSTNESS TO ROTATIONS WITHOUT TRAINING AUGMENTATION

We first train all the networks on MNIST and Fashion MNIST with no rotation augmentation and test against inputs augmented with varying range of random rotations from $\pm 0°$ to $\pm 180°$. Figure 21(top left) for MNIST and Figure 21(top right) for Fashion MNIST shows the performance of these networks to the entire range of rotations. We observe that even though these datasets are small and these networks are designed to be invariant to rotations, their performance drops as the range of random rotations in test data increases. For small range of rotations up to about $\pm 20°$, the performance of these networks on MNIST remains above 95% indicating that MNIST dataset has some natural rotation augmentation. On Fashion MNIST, the performance is above 85% when the test is augmented with small rotations up to $\pm 20°$. When the test is augmented with rotations beyond 25°, the accuracy of these networks drops. We observe that PTN and RotEqNet are more robust than the other networks for MNIST and Fashion MNIST.

On the dataset CIFAR-10 we compare the performance of rotation-equivariant GCNNs with translation-equivariant StdCNNs. This is shown in Figure 21(bottom). GCNNs perform better, as expected, but the performance of both GCNNs and StdCNNs degrades to less than 70% accuracy when the test data is augmented with random rotations larger than $\pm 30°$.

### E.2    ROBUSTNESS WITH TRAINING AUGMENTATION

We train equivariant networks with input augmented with varying range of random rotations from $\pm 0°$ to $\pm 180°$. Figure 22 plots the accuracy for MNIST (top left), Fashion MNIST (top right) and CIFAR-10 (bottom), for both StdCNN's and GCNN's. The red(GCNN) and blue(StdCNN) lines indicate networks for which no train augmentation is done. The $y$ coordinate of a point on the black (resp. brown) line corresponding to $\theta°$ on the X-axis (for $\theta$ in the range of $[0, 180]$) indicates the accuracy of a GCNN (resp. StdCNN) with training data augmented with random rotations in the range $\pm \theta°$ and with test data also augmented with random rotations in the range $\pm \theta°$. For GCNN's we see a gap of about 4-5% on Fashion MNIST and also CIFAR-10, between the accuracies when trained and tested with no rotation augmentations ($\theta = 0$) and when trained and tested with $\pm 180°$ rotations ($\theta = 180$). It is only on MNIST that rotation-equivariant networks achieve almost the same performance ($\approx 1\%$ difference). For StdCNN's the gap is more pronounced, almost 10-15% on Fashion MNIST and CIFAR-10.

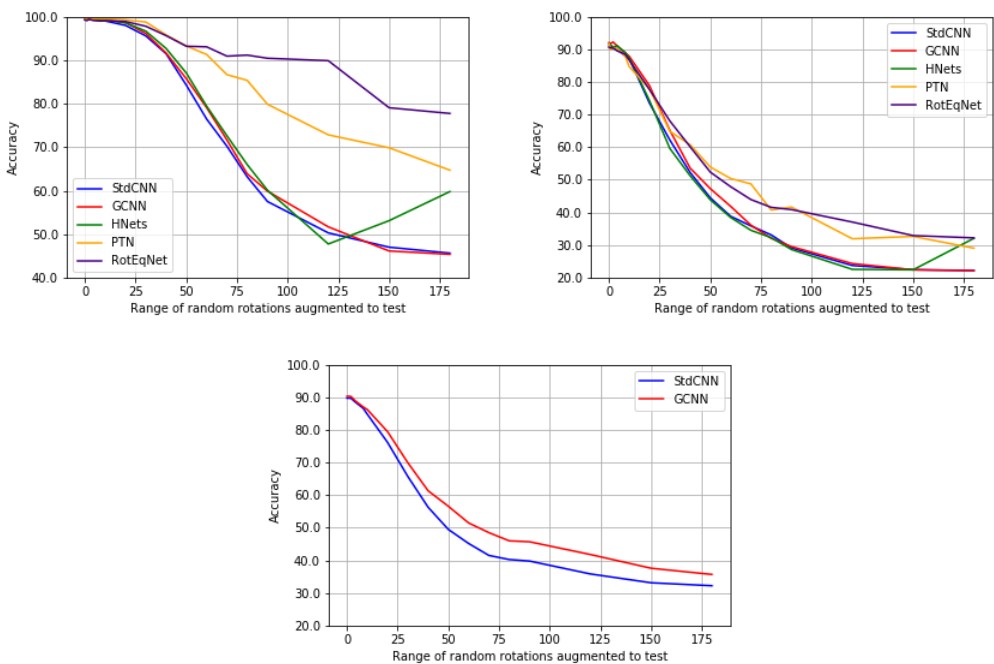

Figure 21: Accuracy of models with no training augmentation. (top left) MNIST, (top right) Fashion MNIST, (bottom) CIFAR-10 .

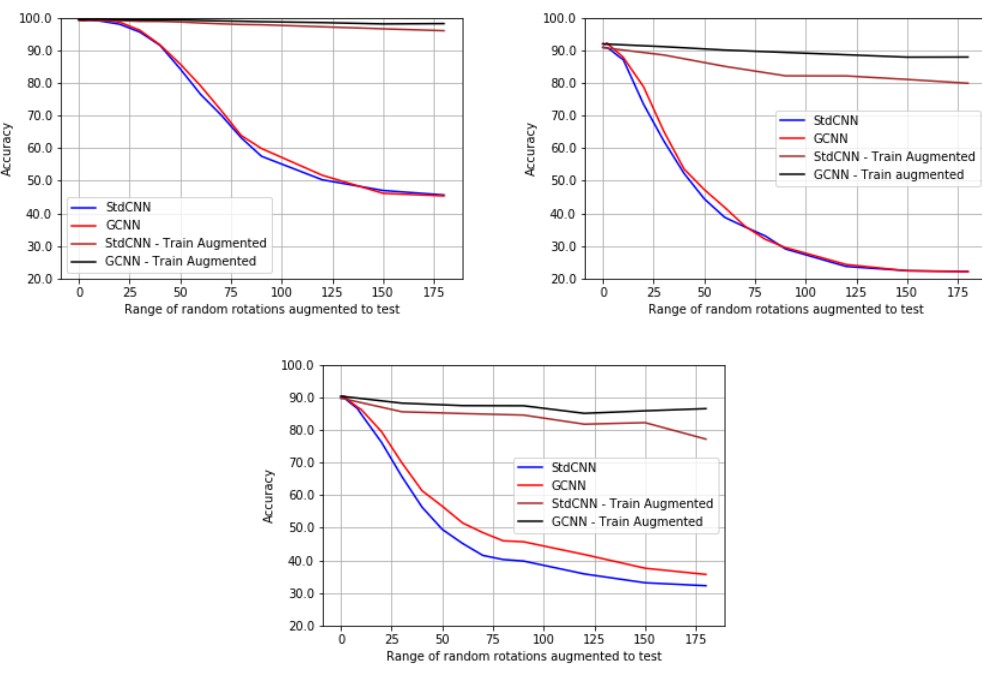

Figure 22: Networks trained with and without augmentation to dataset, random rotation augmentations in $[-\theta^\circ, \theta^\circ]$ range. (top left) MNIST (top right) Fashion MNIST (bottom) CIFAR-10.

### E.3 SAMPLE COMPLEXITY OF NETWORKS

To understand the sample complexity of the networks, we perform two experiments. In the first we train the networks with varying sample sizes of training set and test them on the entire test set. And in the second experiment we do the same as the first with the inputs in train and test augmented with random rotations in the range $[-180°, 180°]$. From Figure 23 for MNIST, Figure 24 for Fashion MNIST and 25 for CIFAR-10, we can see that rotation equivariant networks achieve their best performance safely using $10k$ - $30k$ training samples. This confirms that rotation equivariant networks can do well with smaller training sample size.

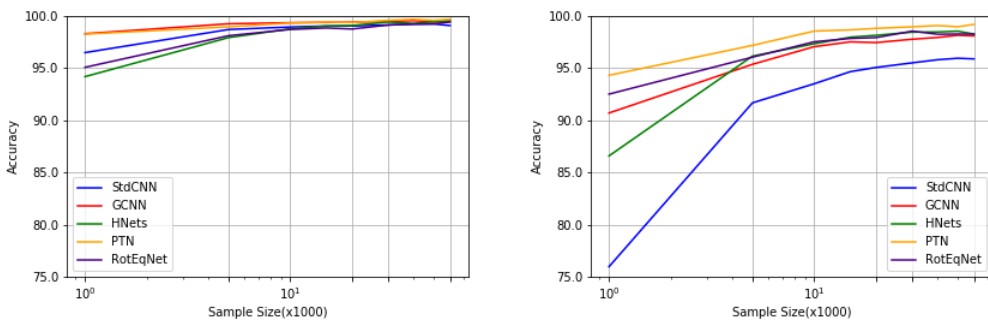

Figure 23: Networks trained with varying training sample size on X-axis. (left) Only MNIST, (right) MNIST train and test augmented with random rotations in $[-180°, 180°]$ range.

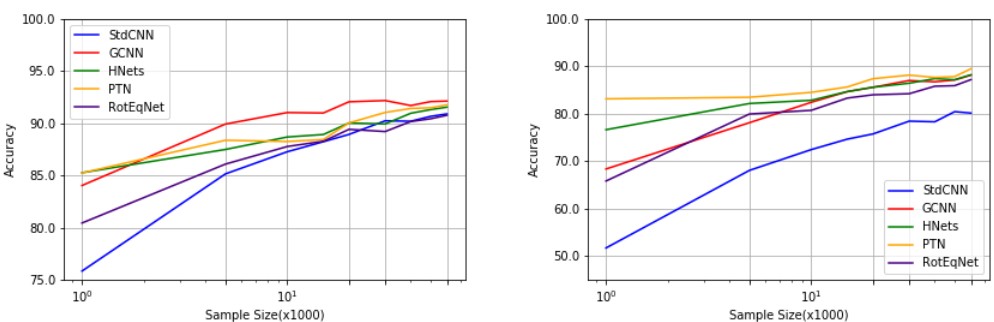

Figure 24: Networks trained with varying training sample size on X-axis. (left) Only Fashion MNIST, (right) Fashion MNIST train and test augmented with random rotations in $[-180°, 180°]$ range.

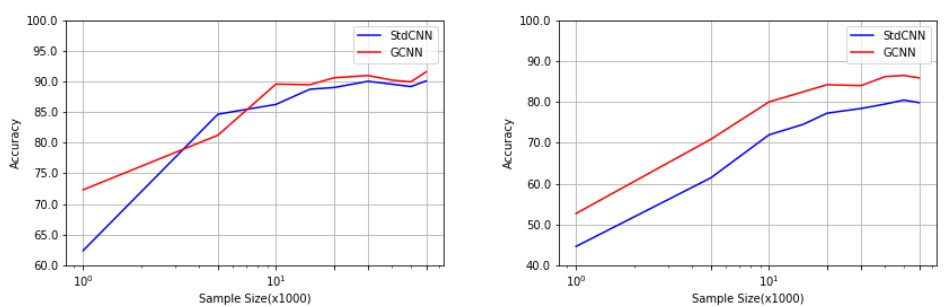

Figure 25: Networks trained with varying training sample size on X-axis. (left) Only CIFAR-10, (right) CIFAR-10 train and test augmented with random rotations in $[-180°, 180°]$ range.

## APPENDIX F   ROBUSTNESS PROFILES USING FOOLING RATE INSTEAD OF ACCURACY

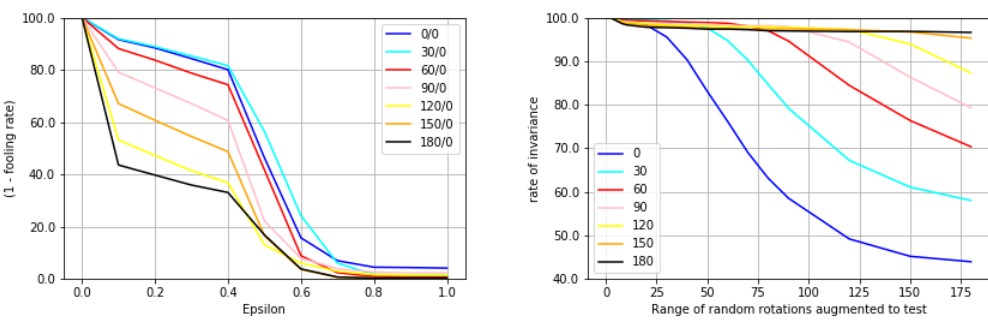

Figure 26: On MNIST, StdCNN trained with varying random rotations in $[-\theta°, \theta°]$ range. (left) Robustness profile, (right) Rotation invariance profile.

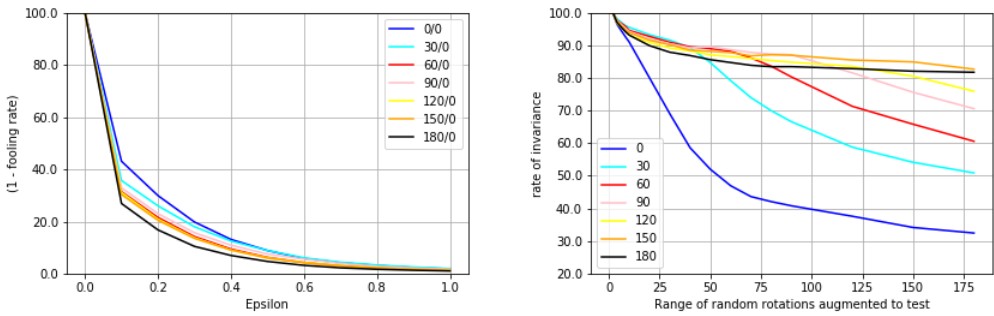

Figure 27: On CIFAR-10, StdCNN trained with varying random rotations in $[-\theta°, \theta°]$ range. (left) Robustness profile, (right) Rotation invariance profile.

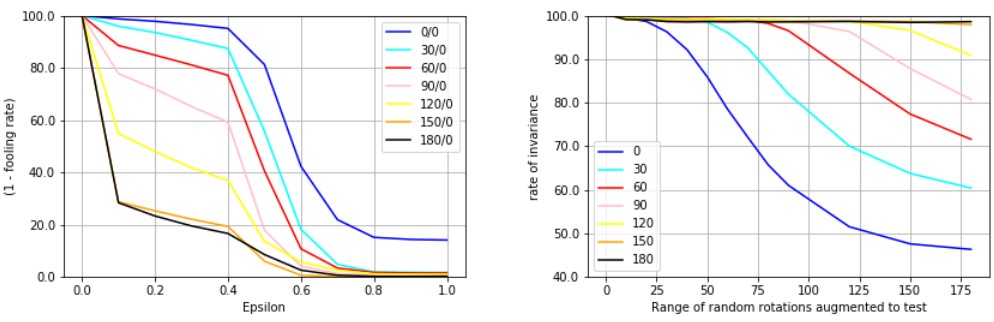

Figure 28: On MNIST, GCNN trained with varying random rotations in $[-\theta°, \theta°]$ range. (left) Robustness profile, (right) Rotation invariance profile.

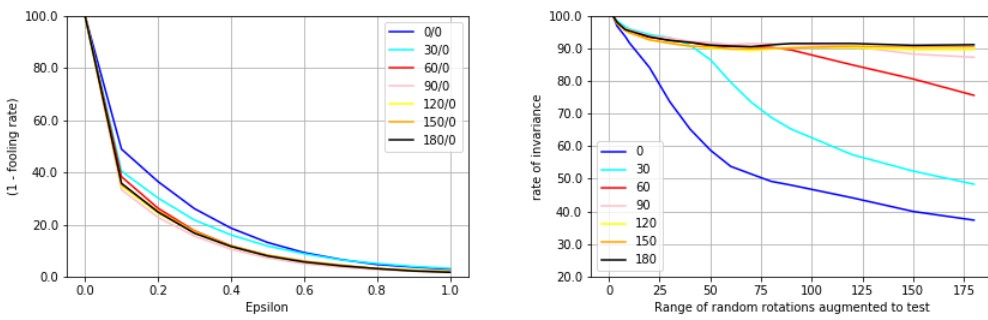

Figure 29: On CIFAR-10, GCNN trained with varying random rotations in $[-\theta°, \theta°]$ range. (left) Robustness profile, (right) Rotation invariance profile.

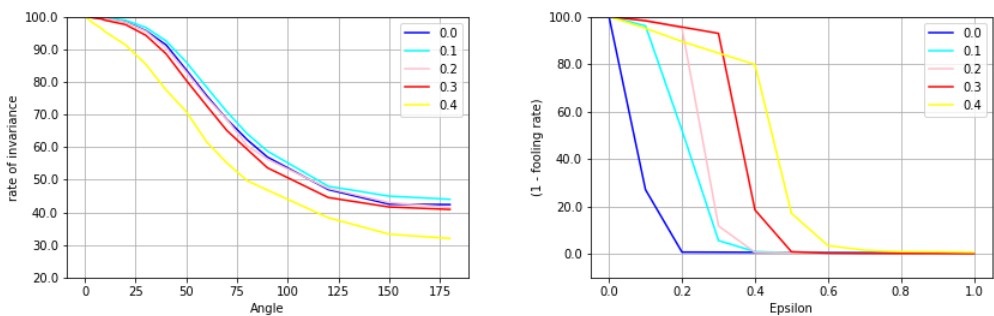

Figure 30: For PGD adversarially trained LeNet based model from Madry et al. (2018) on MNIST (left) Rotation invariance profile, (right) Robustness profile. Different colored lines represent models adversarially trained with different $\ell_\infty$ budgets $\epsilon \in [0, 1]$.

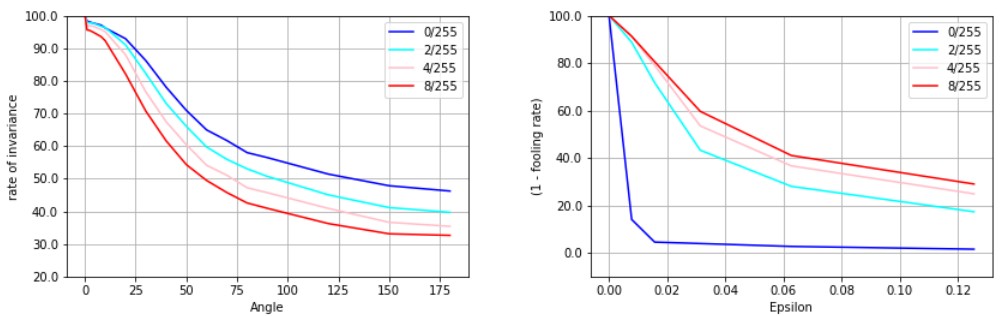

Figure 31: For PGD adversarially trained ResNet based model from Madry et al. (2018) on CIFAR-10 (left) Rotation invariance profile, (right) Robustness profile. Different colored lines represent models adversarially trained with different $\ell_\infty$ budgets $\epsilon \in [0, 1]$.

