# OpenReview forum: "Invariance vs Robustness of Neural Networks"
_ICLR.cc/2020/Conference — Reject_

### Official Review · AnonReviewer1 · 2019-10-22
**Official Blind Review #1**

**Rating:** 3

**Review:**

This paper examines the interplay between the related ideas of invariance and robustness in deep neural network models. Invariance is the notion that small perturbations to an input image (such as rotations or translations) should not change the classification of that image. Robustness is usually taken to be the idea that small perturbations to input images (e.g. noise, whether white or adversarial) should not significantly affect the model's performance. In the context of this paper, robustness is mostly considered in terms of adversarial perturbations that are imperceptible to humans and created to intentionally disrupt a model's accuracy. The results of this investigation suggests that these ideas are mostly unrelated: equivariant models (with architectures designed to encourage the learning of invariances) that are trained with data augmentation whereby input images are given random rotations do not seem to offer any additional adversarial robustness, and similarly using adversarial training to combat adversarial noise does not seem to confer any additional help for learning rotational invariance. (In some cases, these types of training on the one hand seem to make invariance to the other type of perturbations even worse.)

This paper is mostly clear and reasonably written. However, I do not think that the results of this investigation are significant enough to warrant publication at ICLR. In particular, I'm not sure that I really understand the motivation of this research question. I suppose a full notion of robustness would include invariance to perturbations of all types -- whether adversarial or otherwise -- and one might hope that techniques for encouraging such resilience mutually reinforce each other. However, since they are human perceptible, the perturbations associated with invariance are exactly the features that are not associated with adversarial noise, so I don't see why they should be related at all.

From a different perspective, invariance is a property of the true data distribution -- a rotated version of an image of a cat is another valid sample from the underlying distribution -- the invariance property is a type of constraint tying together different elements of the data generating distribution. On the other hand, adversarially perturbed images are often thought to be "off the data manifold" -- i.e. not valid samples from the true underlying distribution. Given this perspective, I am confused about why I should expect to see any interplay between these two ideas. The fact that the authors do not find any interplay is reasonable, but I remain confused about why they were investigating this question in the first place.

Unfortunately, this means that I don't think this work meets the significance criterion for being published at ICLR.

**Experience Assessment:**

I have published one or two papers in this area.

**Review Assessment: Checking Correctness Of Derivations And Theory:**

I assessed the sensibility of the derivations and theory.

**Review Assessment: Checking Correctness Of Experiments:**

I assessed the sensibility of the experiments.

**Review Assessment: Thoroughness In Paper Reading:**

I read the paper at least twice and used my best judgement in assessing the paper.

---

### Official Review · AnonReviewer3 · 2019-10-22
**Official Blind Review #3**

**Rating:** 1

**Review:**

This paper shows empirically that rotational invariance and l infinity robustness are orthogonal to each other in the training procedure. However, the reviewer has the following concerns,

It is already shown in (Engstrom et al., 2017) that models hardened against l infinity-bounded perturbations are still vulnerable to even small, perceptually minor departures from this family, such as small rotations and translations. What is the message beyond that paper that the authors would like to convey?
The experiments are only on MNIST and CIFAR-10.  Training on a larger dataset  like imagenet would make the experiments more convincing.
Going beyond the observation, what shall we do to improve against different perturbation simultaneously? Or is it an impossible task to improve on both?



**Experience Assessment:**

I have read many papers in this area.

**Review Assessment: Checking Correctness Of Derivations And Theory:**

I assessed the sensibility of the derivations and theory.

**Review Assessment: Checking Correctness Of Experiments:**

I assessed the sensibility of the experiments.

**Review Assessment: Thoroughness In Paper Reading:**

I read the paper at least twice and used my best judgement in assessing the paper.

---

### Official Review · AnonReviewer2 · 2019-10-23
**Official Blind Review #2**

**Rating:** 3

**Review:**

This paper analyzes the behaviour of DNN trained with rotated images and adversarial examples. Namely, the paper analyzes the relationship between training with rotated images and the robustness to adversarial perturbations, and vice-versa.

The paper has several technical issues that need to be resolved before drawing any conclusions:

1) “invariance”: this term is not used in the correct way. The fact that the network has the same accuracy when before and after rotation does not mean that the output layer is invariant to rotation. Note invariance in the output layer is a more stringent criterion as it requires that the images get labeled in the same way. The same accuracy can be achieved with completely different labelings of the images. What this paper is evaluation is robustness to rotation vs robustness to adversarial perturbations.

2) It is unclear that Figure 3 is saying that adversarial training does not affect the rotation invariance because there is a general drop of accuracy. The analysis could have been done by evaluating how many images are labelled differently after the rotation, and all the plots will be aligned at 0 degrees.

3) Finding out the robustness to adversarial perturbations is an NP-hard problem. So, for all tested cases in the paper, there could be a perturbation that damaged the model much more than the ones found, which could change the conclusions of the analysis.

4) The networks compared in the two experiments are different networks. There could be a network dependency.

Also, I find the paper poorly written (eg. in the abstract: "Neural networks achieve human-level accuracy on many standard datasets used in image classification.” -> what does it mean “human-level accuracy”?; "The next step is to achieve better generalization to natural (or non-adversarial) perturbations” -> why is this the next step?).


**Experience Assessment:**

I have published in this field for several years.

**Review Assessment: Checking Correctness Of Derivations And Theory:**

N/A

**Review Assessment: Checking Correctness Of Experiments:**

I carefully checked the experiments.

**Review Assessment: Thoroughness In Paper Reading:**

I read the paper at least twice and used my best judgement in assessing the paper.

---

### Decision · Program_Chairs · 2019-12-19

**Decision:**

Reject

**Comment:**

This paper examines the interplay between the related ideas of invariance and robustness in deep neural network models. Invariance is the notion that small perturbations to an input image (such as rotations or translations) should not change the classification of that image. Robustness is usually taken to be the idea that small perturbations to input images (e.g. noise, whether white or adversarial) should not significantly affect the model's performance. In the context of this paper, robustness is mostly considered in terms of adversarial perturbations that are imperceptible to humans and created to intentionally disrupt a model's accuracy. The results of this investigation suggests that these ideas are mostly unrelated: equivariant models (with architectures designed to encourage the learning of invariances) that are trained with data augmentation whereby input images are given random rotations do not seem to offer any additional adversarial robustness, and similarly using adversarial training to combat adversarial noise does not seem to confer any additional help for learning rotational invariance. (In some cases, these types of training on the one hand seem to make invariance to the other type of perturbations even worse.)

Unfortunately, the reviewers do not believe the technical results are of sufficient interest to warrant publication at this time.